# Locality Preserving Property Constrained Contrastive Learning for Object Classification in SAR Imagery

Jing Wang [1], Sirui Tian [1,*], Xiaolin Feng [1], Bo Zhang [2,3], Fan Wu [2,3], Hong Zhang [2,3] and Chao Wang [2,3]

1 School of Electronic and Optical Engineering, Nanjing University of Science and Technology, Nanjing 210094, China; 121104010582@njust.edu.cn (J.W.); fengxiaolin@njust.edu.cn (X.F.)
2 International Research Center of Big Data for Sustainable Development Goals, Beijing 100094, China; zhangbo@radi.ac.cn (B.Z.); wufan@aircas.ac.cn (F.W.); zhanghong@radi.ac.cn (H.Z.); wangchao@radi.ac.cn (C.W.)
3 Key Laboratory of Digital Earth Science, Aerospace Information Research Institute, Chinese Academy of Sciences, Beijing 100094, China
* Correspondence: tiansirui@njust.edu.cn; Tel.: +86-138-0518-9826

**Abstract:** Robust unsupervised feature learning is a critical yet tough task for synthetic aperture radar (SAR) automatic target recognition (ATR) with limited labeled data. The developing contrastive self-supervised learning (CSL) method, which learns informative representations by solving an instance discrimination task, provides a novel method for learning discriminative features from unlabeled SAR images. However, the instance-level contrastive loss can magnify the differences between samples belonging to the same class in the latent feature space. Therefore, CSL can dispel these targets from the same class and affect the downstream classification tasks. In order to address this problem, this paper proposes a novel framework called locality preserving property constrained contrastive learning (LPPCL), which not only learns informative representations of data but also preserves the local similarity property in the latent feature space. In LPPCL, the traditional InfoNCE loss of the CSL models is reformulated in a cross-entropy form where the local similarity of the original data is embedded as pseudo labels. Furthermore, the traditional two-branch CSL architecture is extended to a multi-branch structure, improving the robustness of models trained with limited batch sizes and samples. Finally, the self-attentive pooling module is used to replace the global average pooling layer that is commonly used in most of the standard encoders, which provides an adaptive method for retaining information that benefits downstream tasks during the pooling procedure and significantly improves the performance of the model. Validation and ablation experiments using MSTAR datasets found that the proposed framework outperformed the classic CSL method and achieved state-of-the-art (SOTA) results.

**Keywords:** synthetic aperture radar (SAR); automatic target recognition (ATR); contrastive self-supervised learning (CSL); instance-level contrastive loss; noise-induced estimation of mutual information (InfoNCE); locality preserving projections (LPP)

## 1. Introduction

Synthetic aperture radar (SAR) is a high-resolution coherent imaging radar. It is widely employed in commercial and military surveillance and reconnaissance as an active microwave remote-sensing technology [1]. As the number of operating SAR sensors and images to be interpreted continues to grow, automatic target recognition using SAR imagery (SAR ATR) has piqued the interest of many researchers [2].

Due to the specific electromagnetic imaging mechanism of SAR sensors, it is a considerable challenge to interpret SAR images without domain knowledge and obtain sufficient annotated data for target classification tasks. Recently, with the rapid development of unsupervised or self-supervised deep learning theory, robust representation learning with unlabeled data has received increasing attention from the SAR remote sensing community [3].

Numerous unsupervised models have been developed to mine discriminative features from unlabeled SAR images, including Auto-Encoders (AE) [4], Variational Auto-Encoders (VAE) [5], Generative Adversarial Networks (GANs) [6] and contrastive self-supervised learning (CSL) [7]. Although these methods can adaptively learn robust, informative representations without labels, they still have some issues that greatly deteriorate the performance of their learned features in downstream tasks. AEs and their variants, which learn representation by reconstructing the inputs from the encoded features, focus on the reconstruction performance rather than the discriminative capability of their learned features. Accordingly, there is no guarantee that these models' learned representations will contribute to downstream classification tasks [8]. As a specific AE, VAEs have these same drawbacks. Moreover, they usually assume that the variables follow Gaussian distributions with diagonal covariance matrices, which are quite different from the distributions of SAR images, leading to performance degradation in downstream classification tasks [9]. GANs provide an alternative method of learning the distributions of the inputs through the adversarial procedure with a generator and a discriminator. However, GANs encounter many problems in representation learning tasks related to, for instance, the complexity of their model architecture, the mode collapse problem, the non-convergence and instability of the training procedure, and their requirement for large amounts of data [10].

Recently, the CSL framework, which is a popular form of self-supervised learning, has been adapted to solve various vision tasks, including classification, object detection, and instance segmentation. Using unlabeled data, the CSL models are trained in instance discrimination tasks that discriminate pairs of positive (similar) inputs from a selection of negative (dissimilar) pairs minimizing the InfoNCE loss [11]. CSL models usually obtain a positive sample of the anchor by complex data augmentations and treat the other data in the dataset as negative samples. Due to its excellent performance in unsupervised representation learning, it has been utilized to extract discriminative features from unlabeled SAR images. However, in the absence of true labels, negative samples are often randomly sampled; these samples may have the same label. Furthermore, because SAR images belonging to the same category are highly similar, the CSL framework can dispel samples from the same class using the traditional InfoNCE loss. Both aspects mentioned above lead to a biased representation, greatly affecting these models' performance in downstream classification tasks. To alleviate this problem, a Debiased Contrastive Learning (DCL) [12] model based on the SimCLR [13] framework has been proposed, which exploits a decomposition of the true negative distribution to correct for the sampling of same-label data points. Although the DCL model significantly outperforms the traditional CSL methods, it still struggles to estimate the negative distribution.

To tackle this problem, in this paper, an alternative debiased approach based on the LPP constraint is proposed, called locality preserving property constrained contrastive learning (LPPCL). Similar to the DCL model, the proposed model adopts the SimCLR framework as its prototype due to its tractability and satisfactory results. The proposed method organically combines contrastive loss and the LPP constraint and is able to learn an informative representation of data while simultaneously maintaining the local similarity property in the feature space. Specifically, the LPP constraint is added to the cross-entropy form of the InfoNCE loss as a soft pseudo label, which can introduce additional data structure in order to alleviate the problem of biased sampling and promote performance in representation learning. Moreover, we used a multi-branch structure to produce more training samples through numerous stochastic data augmentations. Finally, we replaced the traditional average pooling approach with self-attention pooling to extract features more effectively.

The main contributions can be summarized as follows:

1.  Prior knowledge of local similarity was embedded into the InfoNCE loss, which was reformulated in the cross-entropy form, reproducing the debiased contrastive loss that the intra-class relationship of nearby samples maintains in the feature space.

2. A multi-branch structure was devised to replace the traditional double-branch structure of CSL, significantly improving the sample diversity, the robustness of representations, and the stability of mutual information estimation.
3. The novel self-attention pooling was introduced to replace the global average pooling in the standard ResNet encoders, providing an adaptive informative feature extraction capability according to the characteristics of inputs and thus avoiding information loss caused by traditional hand-crafted pooling methods.

The rest of this manuscript is organized as follows: a literature review on the CSL method and its application in the SAR ATR is presented in Section 2. A brief summary of backgrounds, including those on LPP and the traditional SimCLR framework, is provided in Section 3, while the proposed new framework is presented in Section 4. Section 5 reports the results of validation and ablation experiments on the MSTAR dataset, with this being followed by the study's conclusion in Section 6.

## 2. Related Work

The CSL approach, which provides a means of enhancing the performance of unsupervised feature learning, is beginning to capture the interest of many researchers. The approach has two essential components: notions of positive and negative pairs of data points. Its training objective, typically the noise-contrastive estimation of mutual information [14], guides the learned representation to map positive pairs to nearby locations in the feature space, as well as negative pairs that are farther apart. Most recent CSL models fall into one of two categories: context-instance contrast (CIC) and instance-instance contrast (IIC) models [15]. CIC models [16,17] aim to represent the affiliation between a sample's local characteristics and its global context representation. However, IIC models analyze the relationships between various samples' instance-level local representations directly. For a variety of classification tasks, instance-level representation is more important than context-level representation, which was first studied in a paper on DeepCluster [18]. The authors suggested using clustering to generate the pseudo-label for every image, and the labels of these images were predicted by a discriminator. Instance discrimination-based approaches, on the other hand, eliminate the sluggish clustering stage and add useful image augmentation mechanisms. InstDisc [19] was a prototype model that used instance discrimination as its pretext task. Based on InstDisc, the MoCo [20] model further investigated the concept of utilizing instance discrimination and significantly increased the number of negative samples by a sample queue. With an online encoder and an offline encoder, it created momentum contrast learning by updating the weights of the offline encoder with the exponential moving average according to the parameters of the online encoder. The momentum contrast learning scheme played a critical role in preventing model collapse. To address the issue of requiring massive negative samples in MoCo, SimCLR [13] used an end-to-end training architecture with a large batch size. SimCLR possessed a two-branch structure that had encoders with shared weights. It also replaced the original linear layer with a projection head to change the dimensions of the representation. Based on SimCLR, Debiased Contrastive Learning (DCL) [12] proposed a debiased contrastive objective that corrected for the sampling of same-label examples without labels. It achieved this by taking the viewpoint of Positive-unlabeled Learning and exploiting a decomposition of the true negative distribution. InfoMin [21] incorporated additional research on enhancing positive samples, it reduced MI between augmented views in order to learn more effective representations. Furthermore, BYOL [22] was devised in a Siamese network framework to reject the need for negative samples in MoCo and SimCLR models. Additional research on SimSiam [23] revealed the significance of the stop-gradient strategy to avoid collapsing in CSL. To conclude, the CSL approach has better generalization performance and fault tolerance in classification tasks.

The developing CSL approach provides a novel method for learning useful features from unlabeled SAR images. However, the traditional contrastive learning method does not perform well on SAR ATR tasks; many efforts have been made to improve the CSL

framework, and several new models have been devised. Wang et al. conducted SAR imagery classification on the basis of pseudo labels and contrastive learning to address the issue of the lack of labeled SAR images [24]. Zhou et al. proposed a limited data loss function (LDLF) that naturally integrated contrastive loss function and cross-entropy loss function. This aimed to solve the problem of overfitting in SAR ATR by combining strong supervision and weak supervision [25]. Based on the instance-level contrastive loss, Zhai et al. proposed batch instance discrimination and feature clustering (BIDFC), which can adjust the embedding distance in the feature space [26]. A contrastive domain adaption methodology was used by Bi et al. to reduce the disparity in distribution between a simulated and actual sparse SAR dataset [27]. The efficiency of multi-view contrastive loss was confirmed by Chen et al. [28]. They suggested the use of a self-supervised method to extract pixel-level feature representations from unlabeled SAR images. To increase the classification accuracy of SAR ATR for ships, Xu et al. modified the SimSiam framework, which is a classic CSL framework, and developed a new positive pair sampling method that considered polarization information [29]. Wang et al. proposed a mixture loss method consisting of contrastive loss and label propagation to investigate the global and local representations in SAR images [30]. Ren et al. proposed a Siamese feature embedding network and leveraged the CSL approach to train a low-dimensional feature space for feature extraction in SAR ATR [31]. The efficiency of CSL-based pre-training models for SAR and optical imagery classification was examined by Liu et al. They used registered examples that had structural consistency as contrastive pairs in the Siamese framework to acquire shared representations of SAR images [32]. For the task of SAR image scene classification, Xiao et al. forwarded a lightweight CSL framework to learn features by maximizing the similarity between augmented views [33]. Liu et al. proposed a clustering-based CSL model in order to map SAR images from pixel space to feature space, promoting node representation and information propagation in the network [34]. Additionally, Yang et al. forwarded a coarse-to-fine CSL framework to extract valuable representations for SAR image classification tasks at the pixel level [35]. Despite these advances, the CSL model in the SAR image processing domain does not perform as well as in the CV domain due to the lack of prior knowledge and biased sampling of negative samples.

## 3. Background

### 3.1. Locality Preserving Projections

The LPP algorithm [36] is a manifold learning strategy that maps high-dimensional data into a low-dimensional manifold where the local relationship of the data is preserved, utilizing the nearest-neighbor graph of the input data. The algorithm's objective function is shown below:

$$\min \sum_{i,j} (e_i - e_j)^2 S_{ij} \tag{1}$$

where $e_i = G^T x_i$, $x_i$ denotes the original samples, $G$ is the optimal projection matrix. The affinity matrix $S$ represents the similarity between $x_i$ and $x_j$, which is established by calculating the cosine distance between two points:

$$S_{ij} = \begin{cases} \frac{x_i^T x_j}{\|x_i\| \cdot \|x_j\|}, & \text{if } x_j \text{ is } k - \text{nearest neighbor of } x_i \\ 0, & \text{otherwise} \end{cases} \tag{2}$$

It is not difficult to determine from (1) and (2) that when the neighboring points $x_i$ and $x_j$ are mapped far away from each other, the loss function with the symmetric weights $S_{ij} = S_{ji}$ will incur a heavy penalty. Therefore, the purpose of minimizing this function is to ensure that the corresponding points will still be neighbors in the low-dimensional projection space if two points are considered to be close by in the high-dimensional space. To some extent, LPP optimally preserves the local neighborhood information of data.

### 3.2. Contrastive Learning and InfoNCE Loss

As a very popular self-supervised learning technique, CSL usually obtains effective image features from the multi-view perspective. In the CSL framework, two views of the input are generated independently by various image transformations. Based on the assumption that each view contains all the required information for the downstream tasks, the CSL network utilizes a dual-branch architecture to learn discriminative representations via an instance discrimination criterion named InfoNCE [11]. Accordingly, the features extracted from different views of the same instance are attracted together, while features extracted from the views of different instances are dispelled.

The SimCLR framework, a crucial baseline for current CSL methods, has a two-branch structure with shared weights in both the encoders and the projection heads of each branch. Instead of using a momentum encoder to update the memory bank, the model requires the batch size of the input data to be large enough to eliminate the requirement for a memory bank. The framework also uses more complicated data augmentation strategies and replaces the original linear layer with a projection head to change the dimensions of a given representation to enhance the effectiveness of contrastive learning. SimCLR is nearly the simplest model, and yet one of the models that achieved the SOTA results in various downstream tasks, including classification, semantic segmentation, and object detection. As shown in Figure 1, the core idea of SimCLR is to learn informative representations by solving an instance discrimination task that pulls together the positive samples from the same instance and pushes away the other $2(N-1)$ negative samples from different instances in the feature space. More specifically, positive samples are generally derived from various augmented images of the same instance, while negative samples are augmented data from different instances. The objective function of SimCLR is derived from the estimation of mutual information (MI) between the inputs and the learned representations. Intuitively speaking, a useful feature should contain as much input information as possible. To this end, an effective strategy is to maximize the MI between embedding features and the original image. However, it is infeasible to compute MI directly in most conditions. An alternative approach is to calculate the upper bound or low bound of MI as an approximal estimation. Recent work has introduced variational bounds with deep learning by using a variational distribution, $p_v(y|x)$, instead of the conditional distribution, $p_c(y|x)$, when calculating the lower bound of the MI:

$$I(X;Y) \geq \mathrm{E}_{p_c(x,y)}[\log p_v(y|x)] + g(X) \tag{3}$$

where, $X$ and $Y$ are random variables in the data space and the learned embedding space, respectively; $x$ represents the samples of $X$; $y$ is the corresponding representation that has maximal MI with $x$ subject to constraints on the mapping. Poole et al. combined the above unnormalized bound with multiple examples [11]. Then, the low bound of the MI can be maximized by minimizing the InfoNCE loss [37]:

$$\mathcal{L}_{InfoNCE} = \mathbb{E}\left[-log \frac{exp(sim(x_i, y_i)/\tau)}{\sum_{k=1}^{N} exp(sim(x_i, y_k)/\tau)}\right] \tag{4}$$

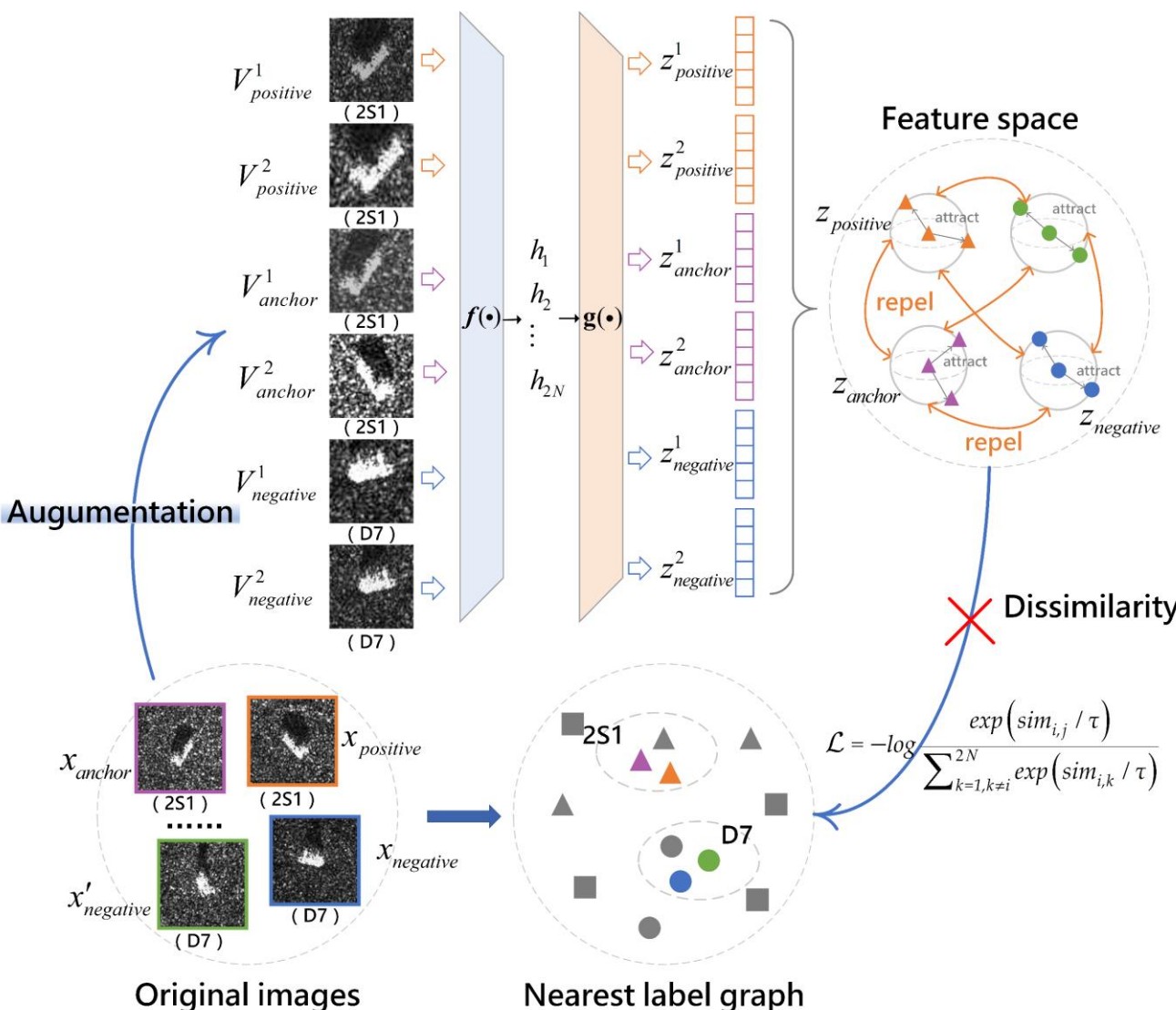

**Figure 1.** Basic concept of the SimCLR framework. $V^1_{anchor}$ and $V^2_{anchor}$ are two augmented views generated by the same anchor instance $x_{anchor}$, which are regarded as a positive pair by SimCLR, while all other views from different instances are regarded as negative samples, whether from the same class like $V^1_{positive}$ and $V^2_{positive}$ and from different class like $V^1_{negative}$ and $V^2_{negative}$, which is also the reason for biased sampling.

The loss function of SimCLR can be obtained by substituting the parameters into Equation (4):

$$\mathcal{L} = -log \frac{exp(sim_{i,j}/\tau)}{\sum_{k=1,k\neq i}^{2N} exp(sim_{i,k}/\tau)} \tag{5}$$

with

$$sim_{i,j} = \frac{z_i^{\mathrm{T}} z_j}{(\| z_i \| \cdot \| z_j \|)} \tag{6}$$

where $z = g_\varphi(f_\theta(V))$, and the positive pair $(z_i, z_j)$ represents the corresponding representation projection of the augmented views $V_i$ and $V_j$. Specifically, this involves the data augmentation $t \sim \mathcal{T}$, $t' \sim \mathcal{T}$ that transforms the given example $x_{anchor}$, resulting in two corresponding views of the same example, denoted $V^1_{anchor}$ and $V^2_{anchor}$ (see Figure 1). Further, $f_\theta(\cdot)$ is the encoder network and $g_\varphi(\cdot)$ represents the projection head. The temperature coefficient $\tau$ is used to modulate the intensity of contrastive learning, which is dependent

on confidence. Larger $\tau$ values result in a smoother probability distribution. By optimizing the objective function in (5), the SimCLR model can be easily trained.

In fact, by combining the positive sample $z_j$ and negative samples $\left\{z_1, z_2 \ldots z_{2(N-1)}\right\}$ together as $\left\{z'_1, z'_2 \ldots z'_{2N-1}\right\} \triangleq \left\{z_j, z_1, z_2 \ldots z_{2(N-1)}\right\}$, the objective function of the SimCLR in (5) can be reformulated as a cross-entropy loss function with one-hot labels:

$$\mathcal{L} = -\Sigma_{q=1}^{2N-1} y_q \log p_q \tag{7}$$

with

$$
\begin{aligned}
p_q &= \frac{exp\left(S'_{i,q}/\tau\right)}{\Sigma_{k=1}^{2N-1} exp\left(S'_{i,k}/\tau\right)} \\
y_q &= \begin{cases} 1, q = 1 \\ 0, otherwise \end{cases} \\
S'_{i,q} &= \frac{z_i{}^T z'_q}{\left(\|z_i\| \cdot \|z'_q\|\right)}
\end{aligned}
\tag{8}
$$

where $\left[p_1, p_2 \ldots p_q \ldots p_{2N-1}\right]$ represents the prediction probability vector and $\left[y_1, y_2 \ldots y_q \ldots y_{2N-1}\right]$ is the corresponding one-hot pseudo label.

In a supervised case, $p_q$ represents the probability that the sample belongs to each class and $y_q$ denotes the true label. In unsupervised or self-supervised cases, however, the label $y_q$ is generated at the instance level, i.e., each instance belongs to a separate semantic class. As a result, samples are not clustered according to their categories in the latent space. This issue is more serious when the samples of different categories are very similar, and the distance between individual samples is smaller than the distance between categories. As a result, samples of the same categories and different categories will all be dispelled, thus affecting the framework's classification accuracy in downstream tasks.

## 4. Methodology

### 4.1. Locality Preserving Property Constrained InfoNCE Loss

To solve the problems associated with the instance-level CSL mentioned above, we aim to introduce additional data structure information into our proposed model to assist its representation learning. The problem with instance-based contrastive learning is that it does not take into account the actual distribution of negative examples when sampling the negative samples, and all other samples except the anchors are regarded as negative samples. Such a sampling scheme leads to bias in the learned representations. Some features extracted from samples of the same class are regarded as negative samples, which seriously weakens the classification capability of the learned features. Given this problem, additional categorical information is required to alleviate the biases in the sampling of negative samples. Obviously, in the case of supervised learning, it is easy to introduce the labels as the categorical information to the cross-entropy form of the SimCLR loss function in (7), and thus biases can be eliminated.

However, in unsupervised representation learning, the prior knowledge we can obtain only comes from the distribution of the data itself. DCL [12] alleviates the biased sampling process by assuming the distribution of negative samples, which could be interfered with the improper prior distribution of the negative samples. Recalling the LPP algorithm, the local similarity relationship in the original data space is preserved in feature space since it accurately reflects the inherent clustering structure of the samples. Motivated by the LPP algorithm, in this study, we considered the local similarity of samples in the data space as the prior knowledge to improve the sampling of negative samples to improve the SimCLR InfoNCE loss.

To accomplish this, the adjacency matrix used for the local similarity measure in the LPP algorithm is introduced into the CSL framework, and its loss function in the cross-entropy form is modified. The aim of this is to preserve the local neighborhood relationship of the original data in the learned features. We first construct the LPP affinity matrix

$S_{ij}$ of the original feature of the target image $[x_1, x_2 \ldots x_N]$, which will maintain the local similarity property in the feature space. For a given sample, the data in the original dataset, which has a local similarity relationship, is regarded as the framework's positive sample. Further, the modified contrastive loss in the cross-entropy form is as follows:

$$\mathcal{L} = -\sum_{i=1}^{2N}\sum_{j=1,j\neq i}^{2N} S_{ij} log \frac{exp(z_i \cdot z_j^T / \tau)}{\sum_{k=1,k\neq i}^{2N} exp(z_i \cdot z_k^T / \tau)} \tag{9}$$

$$S_{ij} = \begin{cases} \frac{x_i^T x_j}{\|x_i\| \cdot \|x_j\|}, & \text{if } x_j \text{ is } k-\text{nearest neighbor of } x_i \\ 0, & \text{otherwise} \end{cases} \tag{10}$$

where $S_{ij}$ can be regarded as the soft pseudo label of the cross-entropy loss, and $k$ represents the quantity of defined adjacent similar samples.

The core idea of the proposed locality preserving property-constrained contrastive learning is illustrated in Figure 2. The proposed model has more than one positive sample. This change comes from the introduction of the LPP algorithm's affinity matrix into the SimCLR framework. The positive samples consist of both the augmented pair $V_{anchor}^i$ and $V_{anchor}^j$ of the anchor instance $x_{anchor}$ and the augmented views $V_{positive}^i$ and $V_{positive}^j$ of $x_{anchor}$'s adjacent sample $x_{positive}$. All these adjacent samples not only have a neighborhood relationship with $x_{anchor}$ in the sample space but also quite possibly come from the same class as $x_{anchor}$. Augmented views of the anchor instance and its adjacent samples constitute a larger positive sample set to assist the representation learning. The other views from different classes are considered negative samples, like $V_{negative}^i$ and $V_{negative}^j$. Specifically, the model constructs the affinity matrix of the original images in the whole dataset to find the nearest first $k$ samples that have higher local similarity properties to $x_{anchor}$. The soft pseudo label $S_{ij}$ is introduced when calculating the contrastive loss value between $x_{anchor}$ and its corresponding positive samples. The modified pseudo label alleviates the performance deterioration caused by the biased pseudo labels without true labels and induces the model to learn better feature representation. This leads to the embedding features of similar image instances being closer to each other and the embedding features of dissimilar image instances being dispelled in the feature space.

Furthermore, (9) can be transformed into the following form:

$$\begin{aligned} \mathcal{L} &= -\sum_{i=1}^{2N} S_{ip} log \frac{exp(z_i \cdot z_p^T / \tau)}{\sum_{k=1,k\neq i}^{2N} exp(z_i \cdot z_k^T / \tau)} - \lambda \sum_{i=1}^{2N}\sum_{j=1,j\neq i,p}^{2N} S_{ij} log \frac{exp(z_i \cdot z_j^T / \tau)}{\sum_{k=1,k\neq i}^{2N} exp(z_i \cdot z_k^T / \tau)} \\ &= \underbrace{-\sum_{i=1}^{2N} log \frac{exp(z_i \cdot z_p^T / \tau)}{\sum_{k=1,k\neq i}^{2N} exp(z_i \cdot z_k^T / \tau)}}_{\mathcal{L}_C} \underbrace{-\lambda \sum_{i=1}^{2N}\sum_{j=1,j\neq i,p}^{2N} sim(z_i, z_j) \cdot S_{ij}}_{\mathcal{L}_{LPP}} \end{aligned} \tag{11}$$

where $\mathcal{L}_C$ is standard contrastive loss. $z_p$ is the positive example. $\lambda$ is the weight coefficient; when it is set to 1, the loss function degenerates into (9). $\mathcal{L}_{LPP}$ can be regarded as the objective function of the LPP between different views in the feature space, where the Euclidean distance similarity is replaced by the normalization exponential cosine similarity between different views. Consequently, it demonstrates that the proposed model's objective function is a contrastive loss function constrained by an LPP term.

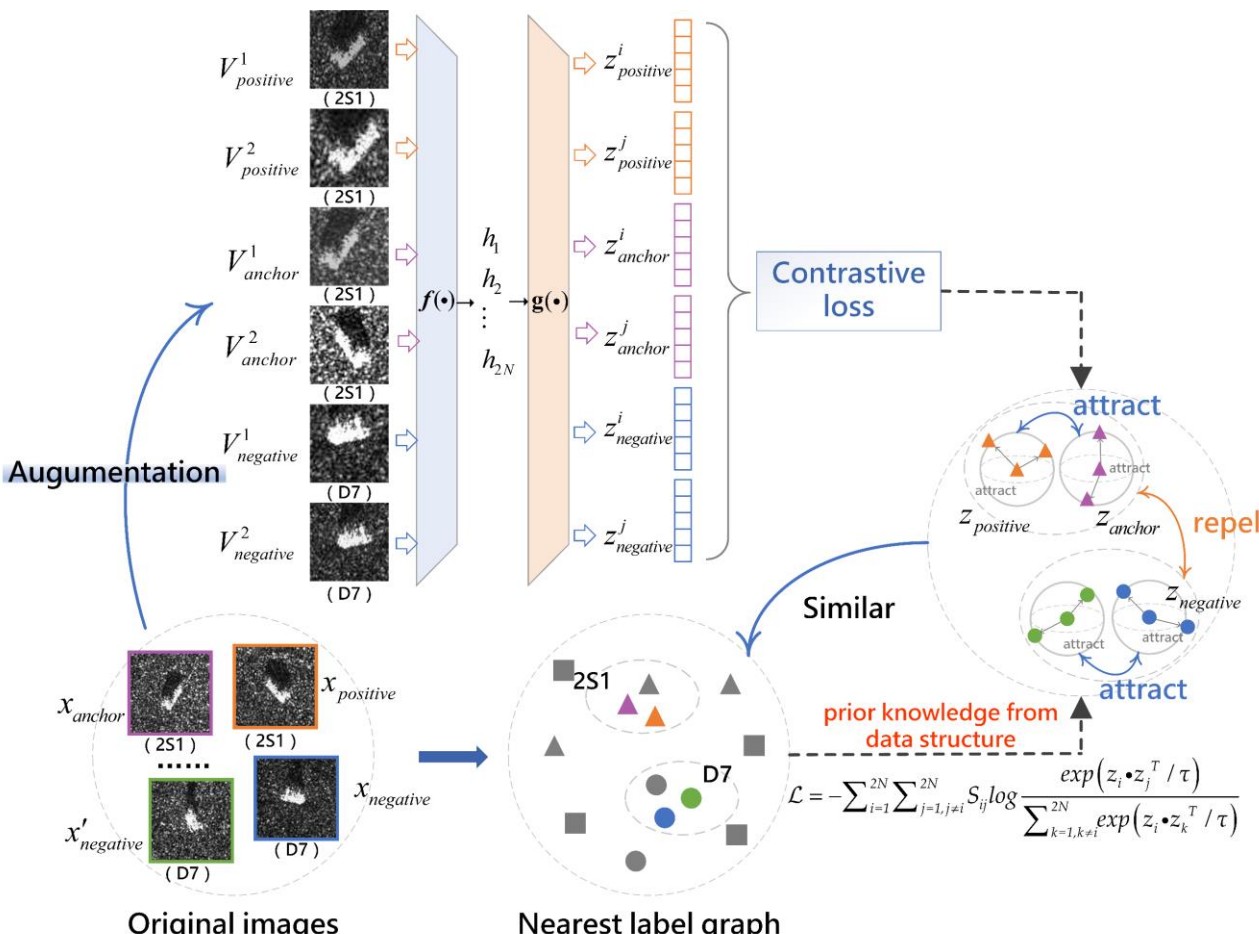

**Figure 2.** Basic architecture of the proposed model. In LPPCL, for augmented view $V_{anchor}^i$, the positive samples include not only augmented view $V_{anchor}^j$ from the anchor instance $x_{anchor}$, but also views $V_{positive}^i$ and $V_{positive}^j$ of $x_{anchor}$'s adjacent sample $x_{positive}$, which quite possibly comes from the same class as $x_{anchor}$.

### 4.2. Multi-Branch Contrastive Learning

In some recent research [12,38], it is found that the robustness of representation learning and the performance of classification tasks could be improved to some extent by increasing the number of positive samples. A similar strategy is therefore included in the proposed model. In order to obtain a robust estimation of mutual information, we adopted a multi-branch structure instead of SimCLR's two-branch structure. As shown in Figure 3, in order to improve the model's representation of augmented views, we augment the original images $[x_1, x_2 \ldots x_N]$ for M(M > 2) times to obtain more views $[V_1^{1,\ldots,m,\ldots M}, V_2^{1,\ldots,m,\ldots M} \ldots V_N^{1,\ldots,m,\ldots M}]$ with $m \in [1, M]$. Obtaining the corresponding representation projection $[z_1^m, z_2^m \ldots z_N^m]$, we then calculate the loss of multiple groups using permutation and combination in pairs. Encoders and projection heads of all branches share their weights. The final loss function value is obtained after the average:

$$\mathcal{L} = \sum_{m,t=1;m\neq t}^{M} \left\{ \sum_{i=1}^{N} \left[ -\log\frac{exp\left(z_i^m \cdot \left(z_i^t\right)^T/\tau\right)}{E_k} - \lambda\sum_{j=1,j\neq i}^{N} S_{ij}\left(\log\frac{exp\left(z_i^m \cdot \left(z_j^m\right)^T/\tau\right)}{E_k} + \log\frac{exp\left(z_i^m \cdot \left(z_j^t\right)^T/\tau\right)}{E_k}\right)\right] \right\}$$
$$E_k = \sum_{k=1,k\neq i}^{N} exp\left(z_i^m \cdot \left(z_k^m\right)^T/\tau\right) + \sum_{k=1}^{N} exp\left(z_i^m \cdot \left(z_k^t\right)^T/\tau\right)$$

(12)

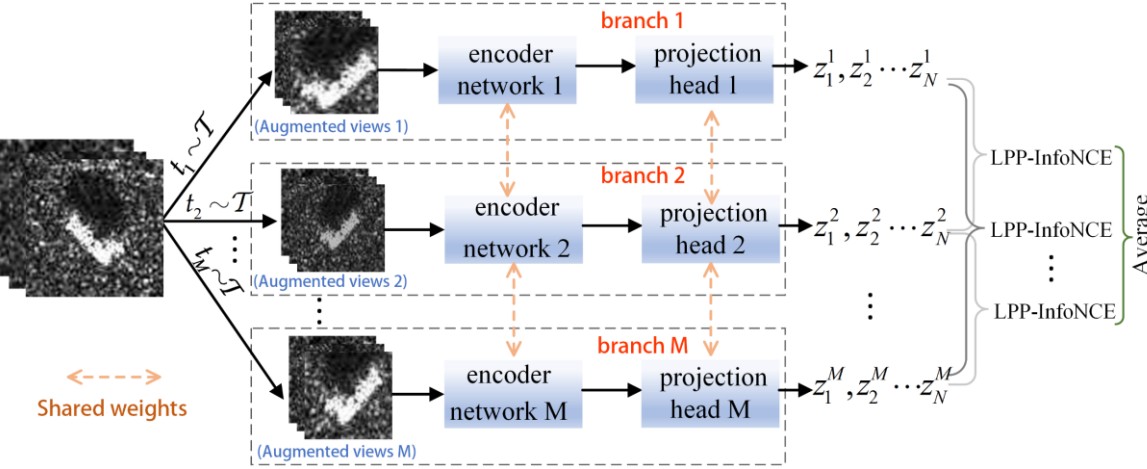

**Figure 3.** The multi-branch structure of the proposed model.

Our experimental results and some recent research [12,38] both demonstrated that the representation learning capability of the CSL models would be boosted by embedding more augmented views into the framework. The multi-branch structure not only enriched the number of positive and negative samples but also increased the diversity of samples, therefore significantly improving the robustness and discriminative capability of the learned features. As shown in Figure 2, in the embedding space, features extracted from different views of samples in the same class are attracted, while features learned from samples of different classes are dispelled so that the local data structure will be preserved. As illustrated in Algorithm 1, the overall loss to optimize the encoder can be formulated as:

---

**Algorithm 1** The proposed LPPCL algorithm

**input:** batch size **N**, matrix $S_{ij}$, constant $\tau$, $m$, the structure of $f$, $g$, $\mathcal{T}$.

**for** $\{x_k\}_{k=1}^{N}$ **do**
  obtain the corresponding affinity matrix $S_{ij}$
  draw M augmentation functions $t_m \sim \mathcal{T}$, $m \in [1, M]$
  **for** $a$ in range (M):
    **for** $b$ in range $(a + 1, M)$:
      **for all** $k \in \{1, \ldots, N\}$ **do**
        $V_k = t_a(x_k), h_k = f(V_k), z_k = g(h_k)$
        $V_{k+N} = t_b(x_k), h_{k+N} = f(V_{k+N}), z_{k+N} = g(h_{k+N})$
      **end for**
      $\mathcal{L}+ = -\sum_{i=1}^{2N} log \frac{exp\left(z_i \cdot z_p{}^T / \tau\right)}{\sum_{k=1, k \neq i}^{2N} exp\left(z_i \cdot z_k{}^T / \tau\right)} - \lambda \sum_{i=1}^{2N} \sum_{j=1, j \neq i, p}^{2N} S_{ij} log \frac{exp\left(z_i \cdot z_j{}^T / \tau\right)}{\sum_{k=1, k \neq i}^{2N} exp\left(z_i \cdot z_k{}^T / \tau\right)}$
    **end for**
  **end for**
  $\mathcal{L} = \frac{\mathcal{L}_{total}}{M(M-1)/2}$
  update $f$ and $g$ to minimize $\mathcal{L}$
**end for**
**return** $f$ and $g$

---

### 4.3. Model Architecture

As shown in Figure 4, the model uses an elaborate network architecture inspired by self-attentive pooling [39]. In the first stage, we make use of a stochastic data augmentation module to transform the selected samples randomly and produce M correlated views $[V_1, V_2 \ldots V_M]$ of the same example. In this study, following the approaches in [12,13,25,26], we use the data augmentation methods comprising random cropping involving resizing, color jitter, random horizontal flip, random grayscale, and Gaussian blur to strengthen the richness of the augmentation combinations and the complexity of the pretext. Secondly,

we adopt the ResNet50 architecture as the encoder network $f_\theta(\cdot)$ to map data into an embedding space, which can extract representation vectors $h = f_\theta(\cdot)$ from augmented samples. A convolutional layer with a $3 \times 3$ kernel is used before the ResNet module to reduce the model's computational complexity. ResNet50 utilizes an average pooling layer at the end of the convolutional layer to generate the feature vectors, which only correspond to one local area on the input feature maps. This pooling technique results in the loss of feature information to some extent. Therefore, we replace the global average pooling in the original ResNet50 network with self-attentive pooling.

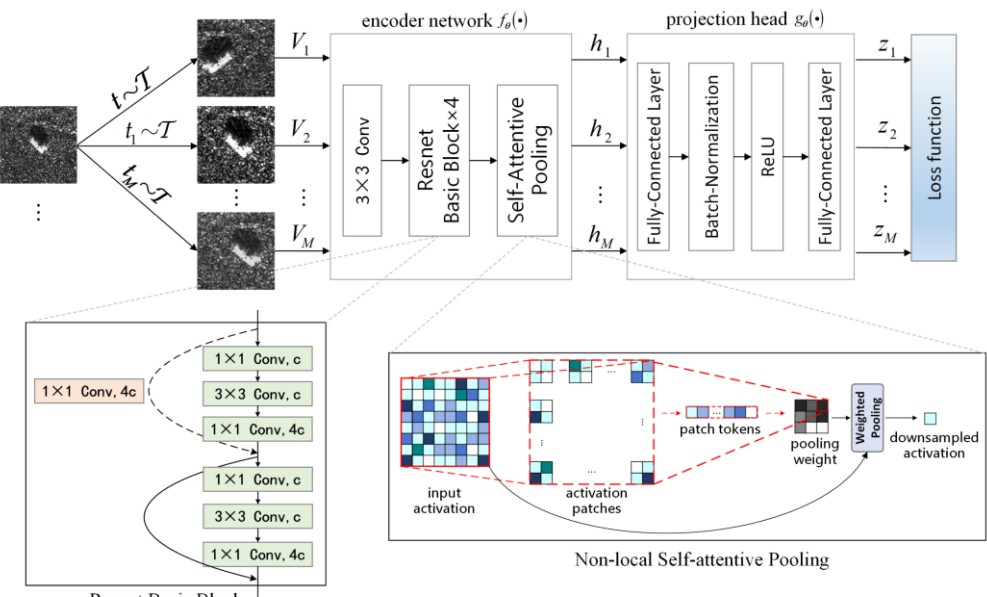

**Figure 4.** The overall framework of the model architecture. This improves the ResNet50 network and uses self-attentive pooling layers after the convolutional layers. The basic block of the Resnet network is shown in the lower left corner, where $c$ denotes the number of channels. The module of the non-local self-attentive pooling is shown in the lower right corner, with the red box shown being the sensitive fields of pooling weights.

Figure 5 shows the unique pooling approach, which uses the multi-head self-attention module and weights the pooling layer using non-local information. At first, the patch embedding module is used to compact and encode local information of the input $x \in \mathbb{R}^{h \times w \times c_x}$, and the output $x_p \in \mathbb{R}^{\left(\frac{h \times w}{\varepsilon_p^2}\right) \times (\varepsilon_r c_x)}$ is a token series, where $\varepsilon_p$ is the patch size and $\varepsilon_r c_x$ denotes the number of output channels. A learnable positional encoding is then added to $x_p$. Secondly, the multi-head self-attention module is employed to simulate the long-term interdependence between patch tokens, yielding a self-attentive token sequence $x_{attn}$. Q, K, and V represent three weight matrices of this module. $x_{attn}$ maintains the size of $x_p$, which is $\left(b \times \frac{h \times w}{\varepsilon_p^2} \times \varepsilon_r c_x\right)$, where $b$ represents the batch size. Thirdly, the spatial and channel information from $x_{attn}$ is decoded in the spatial-channel restoration module, and the $x_{attn}$ is restored to the same size as the input $x$. To be specific, the token sequence is first reshaped to $\left(b \times \varepsilon_r c_x \times \frac{h}{\varepsilon_p} \times \frac{w}{\varepsilon_p}\right)$, and then expanded to the original spatial resolution $(b \times \varepsilon_r c_x \times h \times w)$ by bilinear interpolation up-sampling. The number of channels is afterwards altered back to $(b \times c_x \times h \times w)$ via a convolution process. Finally, the downsampled output feature map $\pi(x)$ from $x_r$ is obtained in the weighted pooling module, which is utilized to produce the whole output activation map. The model's well-designed global view increases its capacity to capture long-term dependencies and aggregation characteristics in order to extract data features more effectively.

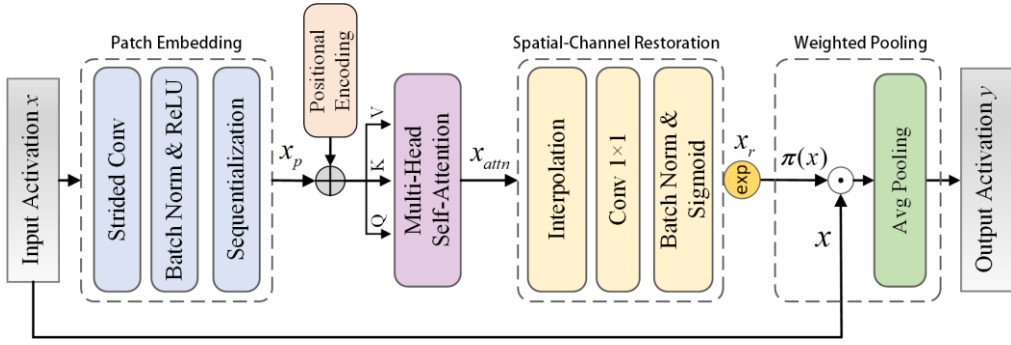

**Figure 5.** Non-local self-attentive pooling.

The representations are then further mapped to the space where contrastive loss is performed using a tiny neural network projection head $g_\varphi(\cdot)$. In this study, we adopt an MLP with two fully connected layers to obtain $z = g_\varphi(f_\theta(x))$. After the training is completed, we utilize the encoder $f_\theta(\cdot)$ and representation $h$ for classification tasks. Although many unsupervised models often utilize a pre-training and fine-tuning scheme, which will fine-tune the representation networks when it is applied to the downstream classification tasks, the proposed model does not require such a scheme. As the proposed model can learn robust and discriminative representations for classification, it is not necessary to apply a fine-tuning process to the trained model to achieve acceptable results in downstream classification tasks.

## 5. Experiments and Results

### 5.1. Experimental Settings

#### 5.1.1. Dataset Description

The MSTAR program provided the experimental dataset required to assess our proposed model. This dataset provides a large amount of SAR images from several kinds of military vehicles as a standard for assessing SAR ATR performance. The photographs and corresponding SAR images are illustrated in Figure 6. Table 1 lists specific details of the experimental dataset.

To thoroughly evaluate the performance of the proposed model, it was tested under two conditions: SOC and EOC. Under SOC, the training set images and the testing set images had the same serial numbers. However, under EOC, images of all the serial numbers were used to test the performance of the proposed method [40]. Following the setup of the EOC experimental method in [41], we used MSTAR datasets with different configurations for evaluation. Due to insufficient training samples, we did not evaluate the performance of the proposed model at the variance of the depression angle. In our experiments, the validations of the other CSL approaches were all carried out using the MSTAR dataset. The training set featured patches acquired at a 17° depression angle, and the testing set was made up of patches taken at a 15° depression angle. Among these, we used the images from BMP2-9563 and T72-132 as training samples for BMP2 and T72.

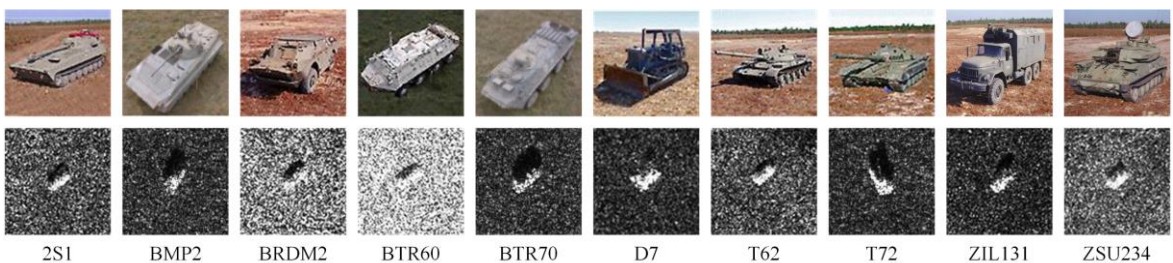

**Figure 6.** Photographs and corresponding SAR images of the MSTAR dataset.

**Table 1.** Detailed information on the MSTAR dataset.

| Type | Serial No. | Number of Samples | |
| --- | --- | --- | --- |
| | | 17° | 15° |
| 2S1 | b01 | 299 | 274 |
| | 9563 | 233 | 194 |
| BMP2 | 9566 | 232 | 196 |
| | c21 | 233 | 196 |
| BRDM2 | E-71 | 298 | 274 |
| BTR60 | K10yt7532 | 256 | 195 |
| BTR70 | c71 | 233 | 196 |
| D7 | 92v13015 | 299 | 274 |
| T62 | A51 | 299 | 273 |
| | 132 | 232 | 196 |
| T72 | 812 | 231 | 195 |
| | s7 | 233 | 191 |
| ZIL131 | E12 | 299 | 274 |
| ZSU234 | d08 | 299 | 274 |

5.1.2. Experiment Setup

To reduce the data volume calculated while ensuring the input data size remained consistent, we used image crop processing based on the center of the target region to resize the input patches to the same shape, which was $64 \times 64$. In addition, many of the target patches had significant variations in target intensity, which obscured the target distinctions. Therefore, intensity normalization was used to map the pixel intensities into the range $[-1, 1]$.

A particular network was used for the proposed method, which is presented in Figure 4 in detail. In this experiment, we set the temperature coefficient $\tau$ to 1 and the number of the adjacent samples $k$ to 5. The balance weight coefficient $\lambda$ of the LPP constraint was set to 1. The branch number M was set to 3 in this study due to the limited memory of the GPU. The preprocessed training dataset was used to train the model, and the Adam optimizer, which had a starting learning rate of 0.0001, was used to optimize it. The exponential decay rates $\beta_1$ and $\beta_2$ were 0.9 and 0.999. The maximum number of iterations was 8000, and the size of the mini-batch was 64. After extracting features from the images, the classification performance was evaluated using a SoftMax classifier. The experiments were implemented on a PC with an Intel(R) Xeon(R) Sliver 4215R CPU @ 3.20 GHz, 256.0 G DDR4, and an NVIDIA GeForce RTX 3090 GPU. The proposed network was built using Pytorch 1.10.2.

The probability of correct classification ($P_{cc}$) serves as a measure to evaluate the performance of the model, which is equal to the number of samples recognized correctly divided by the number of total samples.

*5.2. Recognition Results under SOC*

The experiment conducted under SOC was a standard vehicle target classification problem. Table 2 shows the confusion matrix of the recognition results under SOC. The curve of training loss of the proposed model is shown in Figure 7. The matrix's diagonal lines display the proportion of successfully recognized images for each category. It can be seen that the recognition ratios of BTR60 and T72 were above 94%, the recognition ratios of the other categories were above 97%, and BTR70 attained a 100% recognition rate. The aggregate rate of recognition was 97.94%, which was obviously excellent. Through the structure of the neural network, some robust features were extracted successfully, proving that the proposed model performs excellently under SOC.

**Table 2.** Recognition results under SOC.

| Class | 2S1 | BMP2 | BRDM2 | BTR60 | BTR70 | D7 | T72 | T62 | ZIL131 | ZSU234 | $P_{cc}$ (%) |
|---|---|---|---|---|---|---|---|---|---|---|---|
| 2S1 | 266 | 0 | 0 | 1 | 0 | 0 | 0 | 7 | 0 | 0 | 97.08 |
| BMP2 | 0 | 189 | 0 | 0 | 1 | 0 | 4 | 0 | 0 | 0 | 97.42 |
| BRDM2 | 2 | 0 | 269 | 0 | 0 | 2 | 0 | 0 | 1 | 0 | 98.18 |
| BTR60 | 0 | 0 | 3 | 184 | 0 | 2 | 0 | 5 | 0 | 1 | 94.36 |
| BTR70 | 0 | 0 | 0 | 0 | 196 | 0 | 0 | 0 | 0 | 0 | 1 |
| D7 | 0 | 0 | 0 | 0 | 0 | 272 | 0 | 1 | 0 | 1 | 99.27 |
| T72 | 0 | 8 | 0 | 0 | 0 | 0 | 188 | 0 | 0 | 0 | 95.92 |
| T62 | 1 | 0 | 0 | 1 | 0 | 0 | 0 | 269 | 1 | 1 | 98.53 |
| ZIL131 | 1 | 0 | 0 | 0 | 0 | 0 | 0 | 0 | 270 | 3 | 98.54 |
| ZSU234 | 0 | 0 | 0 | 0 | 0 | 0 | 0 | 1 | 2 | 271 | 98.91 |
| Total | | | | | | | | | | | 97.94 |

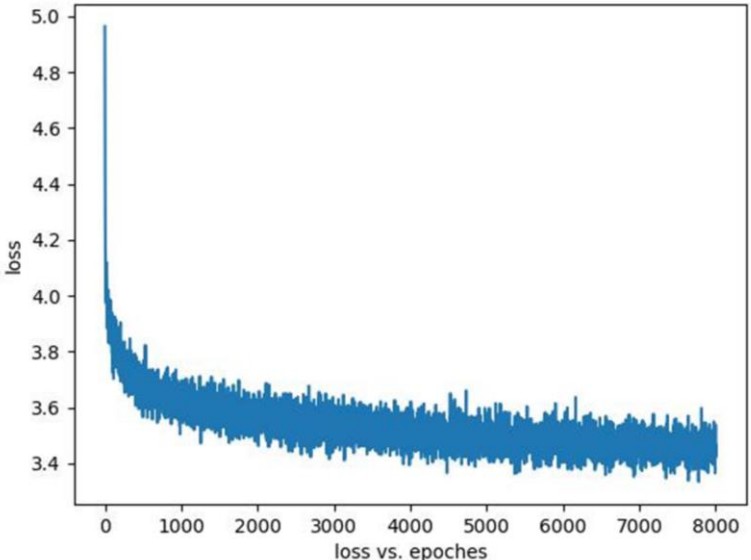

**Figure 7.** The training loss curve of the proposed model.

### 5.3. Recognition Results under EOC

Target recognition was more complicated under EOC due to its variable operational conditions. It was necessary to evaluate the robustness of the proposed model when faced with varying target configurations. Table 3 shows the classification accuracy of the proposed architecture under EOC. The recognition ratios of BMP2, BTR60, and T72 were above 94%, the recognition ratios of other categories were above 97%, and BTR70's recognition rate was 100%. The proposed method obtained a superior overall recognition performance of 97.31%. The recognition results prove that the proposed architecture can recognize targets with different configurations and show resilience to configuration variance.

**Table 3.** Recognition results under EOC.

| Class | 2S1 | BMP2 | BRDM2 | BTR60 | BTR70 | D7 | T72 | T62 | ZIL131 | ZSU234 | $P_{cc}$ (%) |
|---|---|---|---|---|---|---|---|---|---|---|---|
| 2S1 | 266 | 0 | 0 | 1 | 0 | 0 | 0 | 7 | 0 | 0 | 97.08 |
| BMP2 | 0 | 561 | 0 | 0 | 3 | 0 | 22 | 0 | 0 | 0 | 95.73 |
| BRDM2 | 2 | 0 | 269 | 0 | 0 | 2 | 0 | 0 | 1 | 0 | 98.18 |
| BTR60 | 0 | 0 | 3 | 184 | 0 | 2 | 0 | 5 | 0 | 1 | 94.36 |
| BTR70 | 0 | 0 | 0 | 0 | 196 | 0 | 0 | 0 | 0 | 0 | 1 |
| D7 | 0 | 0 | 0 | 0 | 0 | 272 | 0 | 1 | 0 | 1 | 99.27 |
| T72 | 0 | 24 | 0 | 0 | 0 | 0 | 558 | 0 | 0 | 0 | 95.88 |
| T62 | 1 | 0 | 0 | 1 | 0 | 0 | 0 | 269 | 1 | 1 | 98.53 |
| ZIL131 | 1 | 0 | 0 | 0 | 0 | 0 | 0 | 0 | 270 | 3 | 98.54 |
| ZSU234 | 0 | 0 | 0 | 0 | 0 | 0 | 0 | 1 | 2 | 271 | 98.91 |
| Total | | | | | | | | | | | 97.31 |

*5.4. Comparison with Reference Methods*

In this section, the proposed method was compared with SimCLR and DCL. In addition, locality preserving property constrained DCL was also tested in a comparative experiment. When incorporating additional data cluster structure to contrastive loss, we also tried the second different form:

$$\mathcal{L} = \mathcal{L}_C + \lambda' \mathcal{L}'_{LPP} \tag{13}$$

with

$$\mathcal{L}'_{LPP} = \|S_{ij}' - S_{ij}\|$$

$$S_{ij}' = \begin{cases} \frac{z_i{}^T z_j}{\|z_i\| \cdot \|z_j\|}, & \text{if } z_j \text{ is } k - \text{nearest neighbor of } z_i \\ 0, & \text{otherwise} \end{cases} \tag{14}$$

$$S_{ij} = \begin{cases} \frac{x_i{}^T x_j}{\|x_i\| \cdot \|x_j\|}, & \text{if } x_j \text{ is } k - \text{nearest neighbor of } x_i \\ 0, & \text{otherwise} \end{cases}$$

where $\mathcal{L}_C$ represents the contrastive loss in DCL or SimCLR, $\mathcal{L}'_{LPP}$ denotes the distance between the affinity matrix of the corresponding network outputs and the affinity matrix of the original data, and $\lambda'$ is used for balancing $\mathcal{L}_C$ and $\mathcal{L}'_{LPP}$, which we set to 0.1 for the experiment according to the ten-fold cross-validation.

Accordingly, the proposed model was compared with the following models: LPP-constrained SimCLR model in the form of cross entropy (LPPS-I), SimCLR model with a conventional LPP regularization (LPPS-II), LPP-constrained DCL in the form of cross entropy (LPPD-I) and DCL model with a conventional LPP regularization (LPPD-II). We obtained the results for the above methods by implementing these methods with concrete code. In addition, several recently proposed self-supervised learning approaches that achieved state-of-the-art results in SAR ATR were selected to be used in comparison experiments. These included three contrastive self-supervised learning models, i.e., the PL method [24], the CDA method [27], and the ConvT method [30]. In addition, four self-supervised learning methods were used, i.e., the TSDF-N method [9], the ICSGF method [10], the SFAS method [42], and the DKTS-N method [43]. The experimental results for the above methods were obtained from the corresponding references. The recognition performance of various approaches under SOC was then evaluated, as illustrated in Figure 8. Compared with the other approaches, our method recorded the highest $P_{cc}$, i.e., 97.94%; that is, it performed excellently in SAR image recognition. Further, the proposed method yielded features with superior classification performance, even compared with the DCL method, which achieved SOTA results for feature extraction, i.e., 96.74%. The main reason for this is that the proposed structure ensures that the model can learn an informative representation of data but also preserve the local similarity property in the latent feature space, which is overlooked by most CSL methods. The proposed objective function can diminish the influence of instance-level contrastive loss. In addition, applying the LPP affinity matrix constraint yielded better results for SimCLR than for DCL. For the loss function of SimCLR, the form of cross entropy was found to be more conducive to representation learning than the conventional LPP regularization method. However, for the loss function of DCL, the conventional LPP regularization method achieved better performance.

*5.5. Further Analyses of Ablation Study*

For an additional in-depth assessment of whether the proposed model achieves enhanced recognition performance using self-attentive pooling and a multi-branch structure, further studies are discussed in this subsection. Our ablation experiment analyzed the original SimCLR model (SimCLR), the multi-branch SimCLR model (m_SimCLR), and the SimCLR model with self-attentive pooling (SimCLR_p). It also analyzed four cases in which the locality-preserving property constraint was introduced, namely the locality-preserving property constrained original SimCLR learning (LPPCL), the multi-branch LPPS (m_LPPCL), the LPPS with self-attentive pooling (LPPCL_p), and the multi-branch LPPS

with self-attentive pooling (m_LPPCL_p) models. Further, the recognition performance of these models under SOC was compared. Figure 9 displays the recognition rates of each model.

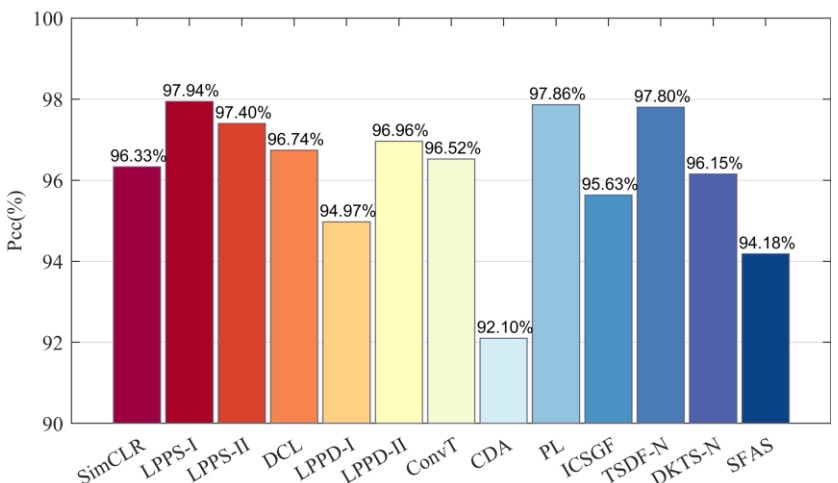

**Figure 8.** Recognition rates of different models.

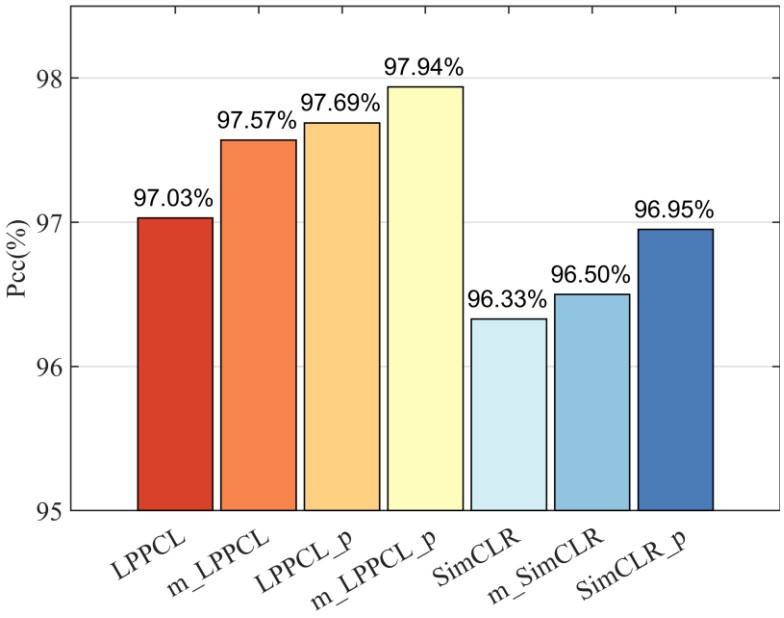

**Figure 9.** Recognition rates of different models.

It can clearly be seen that after introducing the local preserving property constraint to the SimCLR model, its classification accuracy was greatly improved, i.e., its $P_{cc}$ increased from 96.33% to 97.03%. This proves that the introduction of additional local similarity relationships can effectively assist representation learning and alleviate the problem of biased sampling in instance-level contrastive learning. In the feature space, the embedding features of similar image instances were closer to each other, and the embedding features of dissimilar image instances were dispelled effectively. Further, the performance of the multi-branch structure was also better than that of the double-branch structure. The main reason for this is that the multi-branch structure can enrich the diversity of training samples through stochastic data augmentations, thus ensuring the robustness of the model, especially when there are limited samples. It is also beneficial to the calculation of contrastive loss and the estimation of mutual information. By incorporating the multi-branch structure in the traditional SimCLR model, the performance was improved by 0.17%, while

in the LPP-constrained two-branch SimCLR model, the performance was improved by 0.54%. In addition, replacing average pooling with self-attention pooling can assist feature extraction. Unlike average pooling, self-attention pooling can extract features adaptively according to the structure of the data itself and thereby reduce the loss of feature information. Thus, more effective information is retained, which is beneficial to downstream tasks. By incorporating the self-attention pooling in the traditional SimCLR model, the performance was improved by 0.62%, while in the LPP-constrained two-branch SimCLR model, the performance was improved by 0.66%, and in the LPP-constrained multi-branch SimCLR model, the performance was improved by 0.37%. Finally, when the multi-branch structure and self-attention pooling were added together to the SimCLR model with LPP constraint, it recorded the highest $P_{cc}$, i.e., 97.94%, which proves that the proposed strategy can successfully enhance the model's performance.

### 5.6. Experiment with Noise Corruption

In reality, inevitable noise often corrupts measured SAR imagery. In this study, SAR images corrupted by different SNR ratios were used to assess the proposed model's robustness to noise. The original SAR images were considered to be noise-free, which was first transformed into the frequency domain with 2D-IDFT, and additive noises with SNRs ranging from 20 dB to $-20$ dB were applied to the transformed images to produce noise-contaminated images with SNRs specified in (15). The same imaging process was then used to convert the noisy data into the image domain.

$$NR(dB) = 10log_{10}\frac{\sum_{u=0}^{U-1}\sum_{v=0}^{V-1}|f(u,v)|^2}{HW\sigma^2} \tag{15}$$

where $f(u,v)$ is the RCS; $\sigma^2$ is the variance in the additive noise. $H$ and $W$ represent the height and width of the image. Figure 10 shows various noise-contaminated images with varying SNRs. The average experimental results for the proposed model, the SimCLR model, and the DCL model at different SNR levels are shown in Figure 11. As noise interference worsens, the recognition rates of all the models gradually decrease, with the proposed model achieving the highest rate at every SNR level. Even when the image is contaminated by the noise with SNR being 0dB, the geometric and scattering characteristics of the image are not seriously affected by the noise; the $P_{cc}$ of the model is still above 89%, demonstrating the proposed model's resilience under considerable noise interference.

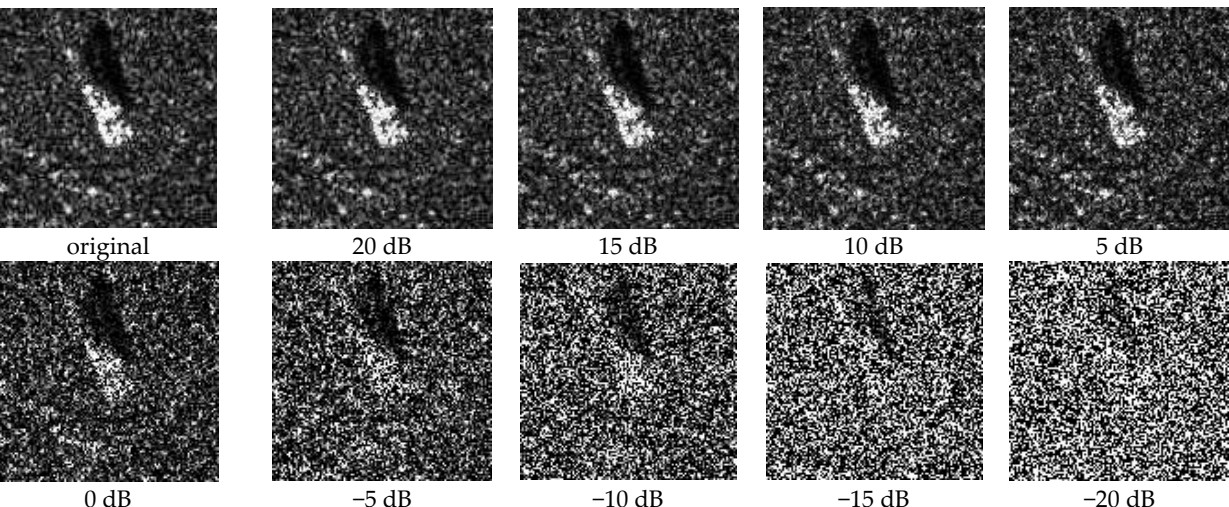

**Figure 10.** MSTAR data with different SNRs.

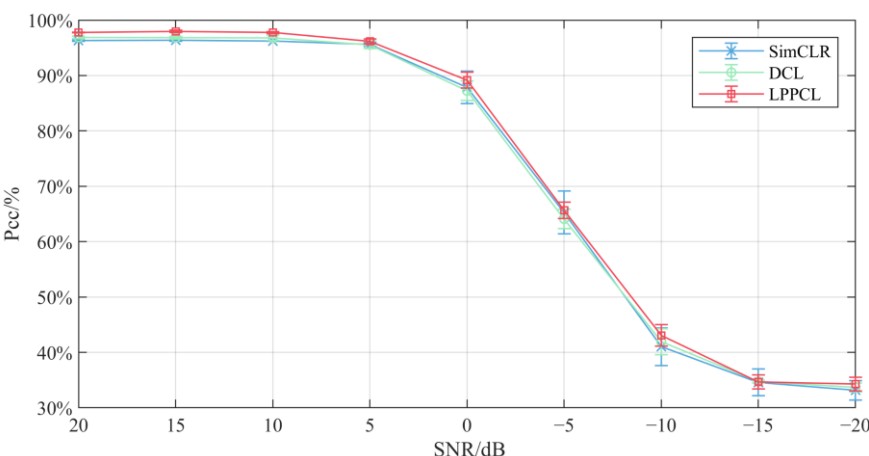

**Figure 11.** Recognition results at different SNR levels.

### 5.7. Experiment with Resolution Variance

Another important consideration when assessing the performance of a model is its robustness to resolution fluctuation. Therefore, we varied the resolution of the SAR images from 0.3 m × 0.3 m to 0.8 m × 0.8 m. Specifically, the sub-band was first retrieved when the spatial SAR images were transformed by 2D-IDFT into the frequency domain. The sub-band data were then resampled in the frequency domain using zero padding before being switched back to the spatial domain. Figure 12 displays several images at various resolutions. In our experiment, the model was trained using the original single-resolution training set, and then the performance of the model was tested on different resolution testing sets. The average experimental results for the proposed model, the SimCLR model, and the DCL model at different resolution levels are shown in Figure 13. It is demonstrated that the proposed model is still effective even with some deterioration in resolution. At all resolutions, the model achieves the highest classification rate when compared with other models. The proposed model's $P_{cc}$ is still higher than 85% even when the resolution is 0.7 m × 0.7 m, proving the model's stability in the face of resolution variation.

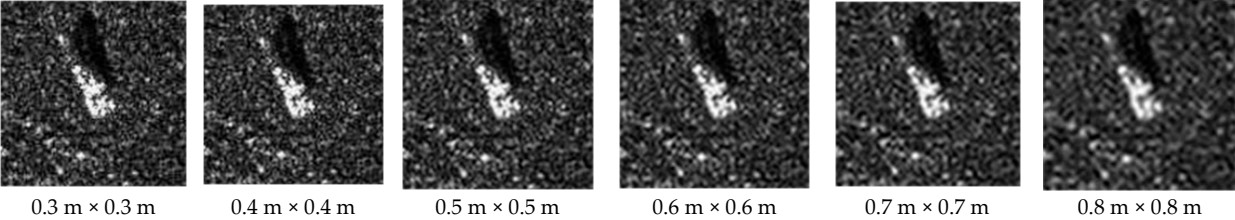

| 0.3 m × 0.3 m | 0.4 m × 0.4 m | 0.5 m × 0.5 m | 0.6 m × 0.6 m | 0.7 m × 0.7 m | 0.8 m × 0.8 m |

**Figure 12.** MSTAR data at different resolutions.

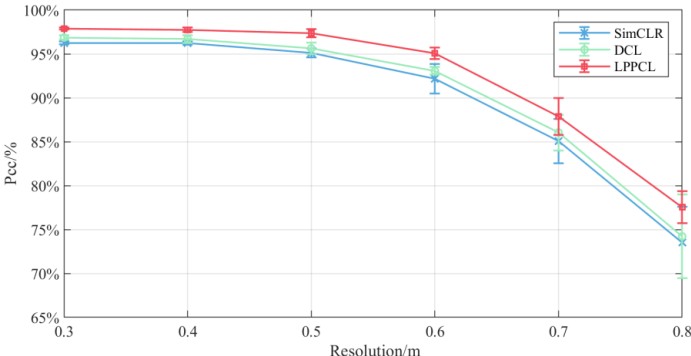

**Figure 13.** Recognition results at different resolutions.

## 6. Conclusions and Future Work

In this paper, a contrastive self-supervised representation learning model is proposed. This model provides an effective method for learning useful features from unlabeled SAR images. Its use of the local preserving property constraint method prevents performance deterioration caused by biased pseudo labels without true labels; rather, the model not only learns informative representations of data but also preserves the local similarity property in the latent feature space. Further, the proposed model's multi-branch structure increases the diversity of training samples, enhancing the model's capacity for representation. In addition, its use of self-attention pooling assists the model with learning informative features, which is conducive to downstream classification tasks. Experiments demonstrated that the proposed model outperformed most of the present CSL algorithms and obtained the SOTA performance without supervised knowledge.

**Author Contributions:** Conceptualization and Validation, J.W. and S.T.; methodology, S.T. and J.W.; software, J.W., F.W. and B.Z.; writing—original draft, J.W. and X.F.; writing—review and editing, C.W. and H.Z.; visualization, J.W. and X.F.; supervision, S.T. and C.W.; project administration, S.T.; funding acquisition, S.T. All authors have read and agreed to the published version of the manuscript.

**Funding:** This research was funded in part by the Key Program of National Natural Science Foundations of China under Grant No. 41930110.

**Data Availability Statement:** Not applicable.

**Acknowledgments:** The authors thank the US Air Force Research Lab for providing the public MSTAR data at: https://www.sdms.afrl.af.mil/index.php?collection=mstar (accessed on 1 September 2022). In addition, we are grateful to the anonymous referees for their instructive comments.

**Conflicts of Interest:** The authors declare no conflict of interest.

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
