# Peer review of "Locality Preserving Property Constrained Contrastive Learning for Object Classification in SAR Imagery"

_remotesensing, doi:10.3390/rs15143697_

Round 1

Reviewer 1 Report

To address the problem of SAR automatic target recognition with limited labeled data, this paper proposes a contrastive learning object classification method based on LPP constraints. This method introduces LPP constraints into the traditional contrastive learning framework, enabling the model to learn clustering relationships between various classes in the latent feature space. The loss function and network structure are optimized, and good performance is finally achieved on the MSTAR dataset. The overall structure of the article is reasonable, and the expression is clear, but there are some questions that the author should carefully consider and answer:

1. Section 2.2 mentions that the CSL structure assumes that the generated multi-view data meets the needs of downstream tasks. Actually, the multi-view data generation method used in this paper comes from SimCLR. How to ensure that these data augmentation methods are suitable for SAR-ATR?

2. Comparative learning methods typically use pre-training & fine-tuning modes. Does the proposed method also require fine-tuning to achieve classification tasks? There is no clear introduction in the paper; please supplement it.

3. The only difference between EOC and SOC in this paper is that BMP2 and T72 have added some samples from different series. This setting is different from the commonly used EOC setting of MSTAR. How is the applicability of the proposed method in scenarios with varying compression angles?

4. There are differences between the symbols used in formula (19) and the corresponding text description. Please check and unify them.

5. Section 4.6 states that "Even when the image was contiguous by the noise with SNR being 0dB, the image's geometric and scattering characteristics were not serially affected by the noise.". The SNR definition method here is based on all pixels in the image, which is different from the SNR definition method in radar signal processing. May cause misunderstandings; it is recommended to optimize.

6. The experimental setup in Section 4.7 needs to be described more clearly. Is it training at a single resolution, multi-resolution testing, or training and testing with data from each resolution separately?

The overall expression of this paper is standardized and fluent. Some details still need further refinement, and it is recommended that the author carefully revise them.

Reviewer 2 Report

This paper presents a locality preserving property constrained contrastive learning for object classification in SAR imagery. The authors made several improvements, including introducing the locality preserving property into contrastive learning, using multiple branches and data augmentation methods, and the self-attention pooling. Generally speaking, the paper is well written and the results are interesting. I only have some minor comments.

1.     Please explain the variables in line 207, what’s x and y?

2.     In Figure. 4, what’s the size of Xattn after the multi-head self-attention? I do not understand the spatial-channel restoration module. Usually Xattn has the same size as the input token. What does interpolation mean? How do you change Xattn into a weight map with the same size as the input x?  

The English expression can be further improved.

Reviewer 3 Report

The paper presents several challenges for readers due to the disorganized presentation of numerous pieces of information, some already well-known in the literature. Consequently, extracting new elements from the content is difficult, and the experimental evidence provided is not entirely convincing. To address these issues, the following improvements are suggested.

I suggest the following to facilitate understanding of the paper and improve its organization.

(a) Passages from the introduction that discuss the state-of-the-art in the literature should be extracted, and a separate section titled "Related Work" should be created. If the current section titled "Related Work" is to be retained (as mentioned later), it should be renamed as "Background" or "Basic Notion." However, any passages in the introduction that point out critical issues in current systems, indicate the gap the authors intend to fill, and highlight the innovativeness of the contribution should be retained or appropriately summarized.

(b)    Subsections 2.1 and 2.2 provide excessive detail on the LPP algorithm and CSL framework. It may be beneficial to refer to relevant sources for much of this information and focus on describing only the modified steps. This will help streamline the content and make it more concise.

(c) Subsection 3.1 should be placed in the section here indicated as "Background" instead of placing it in the Methodology section. The same considerations as in (b) apply to this subsection. Additionally, Figures 1 and 2, which feature cats and dogs, are generic and do not contribute significantly to understanding the paper. These figures are out of context and do not easily correlate with Figure 3. This subsection needs to be significantly changed.

(d)    Subsections 3.2 and 3.3 need to be amended because many methodological aspects are ambiguous, and the description does not highlight the novelty of the method.

Regarding the testing of the method, the following changes should be made:

1.            It is recommended to perform experimental validation of the method on an additional dataset, preferably one with different characteristics from MSTAR. This will provide a more comprehensive evaluation and demonstrate the method's generalizability beyond a single dataset.

2.       The augmentation process applied to the datasets should aim to create balanced training and validation sets. It is important to address this aspect in detail, explaining the specific techniques used to achieve data balance and how it contributes to the method's overall effectiveness.

3.              In order to provide a comprehensive analysis of the results, it is suggested to include accuracy and loss graphs for both the training and validation sets. These graphs should be presented for the proposed and the benchmark methods, allowing for a clear visual comparison of their performance.
